# In Vivo Modulation of Angiogenesis and Immune Response on a Collagen Matrix via Extracorporeal Shockwaves

**DOI:** 10.3390/ijms21207574

**Published:** 2020-10-14

**Authors:** Diana Heimes, Nadine Wiesmann, Jonas Eckrich, Juergen Brieger, Stefan Mattyasovszky, Peter Proff, Manuel Weber, James Deschner, Bilal Al-Nawas, Peer W. Kämmerer

**Affiliations:** 1Department of Oral- and Maxillofacial and Plastic Surgery, University Medical Center Mainz, Augustusplatz 2, 55131 Mainz, Germany; nwiesman@uni-mainz.de (N.W.); bilal.al-nawas@unimedizin-mainz.de (B.A.-N.); peer.kaemmerer@unimedizin-mainz.de (P.W.K.); 2Molecular Tumor Biology, Department of Otorhinolaryngology, Head and Neck Surgery, University Medical Center of the Johannes Gutenberg University, Langenbeckstraße 1, 55131 Mainz, Germany; Jonas.eckrich@unimedizin-mainz.de (J.E.); brieger@uni-mainz.de (J.B.); 3Department of Orthopedics and Traumatology, University Medical Center of the Johannes Gutenberg University of Mainz, Langenbeckstraße 1, 55131 Mainz, Germany; stefan.mattyasovszky@unimedizin-mainz.de; 4Department of Orthodontics, University Hospital Regensburg, Franz-Josef-Strauß-Allee 11, 93053 Regensburg, Germany; peter.proff@ukr.de; 5Department of Oral and Maxillofacial Surgery, Friedrich-Alexander University Erlangen-Nürnberg, 91054 Erlangen, Germany; manuel.weber@uk-erlangen.de; 6Department of Periodontology and Operative Dentistry, University Medical Center of the Johannes Gutenberg University, Augustusplatz 2, 55131 Mainz, Germany; james.deschner@uni-mainz.de

**Keywords:** oral inflammation, angiogenesis, extracorporeal shockwave therapy, collagen matrix, mucoderm^®^, chorioallantoic membrane assay, macrophage response, vascular endothelial growth factor, matrix metalloproteases

## Abstract

The effective management of tissue integration and immunological responses to transplants decisively co-determines the success of soft and hard tissue reconstruction. The aim of this in vivo study was to evaluate the eligibility of extracorporeal shock wave therapy (ESWT) with respect to its ability to modulate angiogenesis and immune response to a collagen matrix (CM) for tissue engineering in the chorioallantoic membrane (CAM) assay, which is performed with fertilized chicken eggs. CM were placed on the CAM on embryonic development day (EDD) 7; at EDD-10, ESWT was conducted at 0.12 mJ/mm^2^ with 500 impulses each. One and four days later, angiogenesis represented by vascularized area, vessel density, and vessel junctions as well as HIF-1α and VEGF gene expression were evaluated. Furthermore, immune response (iNOS2, MMP-9, and MMP-13 via qPCR) was assessed and compared between ESWT- and non-ESWT-groups. At EDD-14, the vascularized area (+115% vs. +26%) and the increase in vessel junctions (+751% vs. +363%) were significantly higher in the ESWT-group. ESWT significantly increased MMP-9 gene expression at EDD-11 and significantly decreased MMP-13 gene expression at EDD-14 as compared to the controls. Using the CAM assay, an enhanced angiogenesis and neovascularization in CM after ESWT were observed. Furthermore, ESWT could reduce the inflammatory activity after a latency of four days.

## 1. Introduction

Tissue engineering in the maxillofacial region aims for regeneration of soft and hard tissue deficiencies in order to sustain oral-related health as well as to improve masticatory function. As autograft harvesting results in increased local morbidity, allogeneic, alloplastic, and xenogeneic grafts are frequently used, even though autogenous tissue remains the biological gold standard [1,2,3]. Most studies in biomaterial research mainly focus on the mechanical and functional properties of the material [4]. However, structural integration of biomaterials including (neo-) angiogenesis/vascularization of the transplants as well as modulation of the immune response at the host site, are also crucial for success [5]. Xenogeneic collagen matrices (CM) are used for soft tissue as well as hard tissue regeneration purposes, being either barrier or remodeling matrices [3,6]. Experimental findings suggest that CM may also have an active role in promoting tissue regeneration [7,8]. For example, CM may have an additional influence on graft vascularization, which is known to have a pivotal role for successful tissue integration [9,10]. However, it was previously shown that acellular dermal CM induce connective tissue formation resulting in scarring and contraction. These authors suggested that prolonged inflammatory processes induced fibrosis [11]. The balance between pro- and anti-inflammatory activity ideally leads to new tissue formation and a degradation of the biomaterial. This immunological interplay, especially regarding different types of macrophages, was shown to modulate the foreign body reaction on CM [4,12].

Extracorporeal shockwave therapy (ESWT) is defined as a biphasic change of pressure during a short period of time with a positive wave of high intensity (40 MPa, ~1 μs duration) and a negative wave of low intensity (10 MPa, ~3 μs duration) [13]. It was first used to treat kidney stones [14]; today, radial and focused ESWT are FDA-approved therapies for the treatment of musculoskeletal disorders and a variety of off-label applications [15]. Radial shockwaves are generated by firing a projectile within a guiding tube that strikes a metal applicator placed on the patients’ skin. After reaching the tissue boundary, they spread non-invasively into the tissue [16] and transform into mechanical energy. The biological effect is directly proportional to the impedance difference between the two adjacent media [13]. The exact mechanisms by which shockwaves affect the tissue metabolism are still unknown. One possibility is the cavitation phenomenon, which is defined as the rapid formation, expansion, and collapse of ultrastructural bubbles in liquids induced by pressure changes [16]. It is thought to induce an asymmetrical fluid stream within the tissue [13]. Furthermore, it was suggested that membrane hyperpolarization and the formation of free radicals may be sources of the biological effects [15]. Several studies could show a positive effect of ESWT on tissue regeneration and cellular proliferation [14,17,18,19,20]. In a rat model, VEGF was significantly increased in hard and soft tissue samples after ESWT [15,21]. HIF-1α levels were significantly elevated in osteoblasts after in vitro stimulation [22] and ESWT-stimulated tissues showed higher vascularization both in animal and human trials [22,23,24,25]. Furthermore, ESWT was able to affect cellular inflammation by down-regulating leukocyte infiltration [26] and via modulation of macrophage activity [27]. By inhibition of NF-κB-associated proteins, such as iNOS, TNF-α, ICAM, VCAM, and COX-2, the local inflammatory response was shown to be affected by ESWT on the molecular level [28]. As already stated, the balance between pro- and anti-inflammatory action is crucial for successful integration of a biomaterial [4]. A long lasting M1 activity results in tissue damage, whereas the tissue-reparative and vasculogenic activity of M2 macrophages is required [29]. It could further be shown that a hyperfunction of M2 macrophages impairs a constructive remodeling resulting in scarring and fibrosis [1].

The extraembryonic chorioallantoic membrane (CAM) is a high vascularized, non-innervated tissue of the developing chicken embryo. It allows in vivo analysis of angiogenesis and immunoreaction on biomaterials within a living organism with a minimum of stress and suffering of the experimental animal [30]. The CAM assay is deemed to be the bridge between in vitro and conventional rodent in vivo animal models, especially as the inflammatory response is similar to that observed in mammalian models [31]. Via a small window at the upper part of the eggshell, the CAM can be accessed directly. Biomaterials can be placed directly onto the CAM and analyzed longitudinally regarding angiogenesis and neovascularization as well as for immunoreactions as an endpoint measurement after extraction of the transplant. Macrophage-like cells appear at embryonic development day (EDD) two to four and they numerically increase during the following development [32,33]. They show typically phagocytotic activity after cell death suggesting homology to mammalian fetal macrophages. Moreover, they show anti-microbial functions such as oxidative burst, NO-production, lysozyme, and cytokine expression [33]. Even though the exact cell migration process is largely unexplored in avian species, similarities to murine and human models are assumed, especially regarding an enhanced iNOS activity after contact of macrophages to LPS and interferon gamma [33]. Considering the great advantages of this experimental in vivo model combined with large homologies between the avian and the mammalian immune system, the CAM assay was chosen as the experimental system.

The aim of the study was to test in vivo whether ESWT is able to modulate the foreign body response towards a xenogeneic collagen membrane with respect to the promotion of vascularization and the suppression of exaggerated immunological reactions during integration of this biomaterial as a new and promising path in tissue engineering.

## 2. Results

### 2.1. Intravital Fluorescence Microscopy

#### 2.1.1. Determination of the Vascularized Area on the Collagen Matrix

Quantitative analysis of the vascularized area was performed by measuring the invasion depth of blood vessels and connective tissue into the collagen matrix using intravital fluorescence microscopy. Multiple image arrays were taken at EDD-11 and EDD-14 (one day and four days after ESWT-/sham-treatment) in order to track vessel development over time (see Figure 1).

One day after ESWT treatment (EDD-11), the analysis could show a significantly higher coverage of the transplant by vasculature within the control group (−ESWT: 44.47%, 95%-CI [34.99–53.95] vs. +ESWT: 35.88%, 95%-CI [30.01–41.75]; *p* = 0.03). Four days after treatment, the percentage of the vascularized area was shown to be significantly higher in the ESWT group (+ESWT: 75.06%, 95%-CI [45.53–68.83] vs. −ESWT: 57.19%, 95%-CI [67.53–82.59]; *p* = 0.006) (see Figure 2a). To assess the longitudinal development, we further analyzed the percentage increase of the vascularized area on the transplant between EDD-11 and EDD-14. The statistical analysis could show a significantly higher increase of the vascularized area within the ESWT-treated (114.49%, 95%-CI [54.33–174.65]) compared to the control group (25.59%, 95%-CI [5.74–45.45]; *p* = 0.004) (see Figure 2b), which leads to the assumption that ESWT was able to increase the vascularization in porcine-derived CM.

#### 2.1.2. Vessel Density Analysis

A vessel density analysis was performed to further test the hypothesis of a potential pro-angiogenetic effect of ESWT on dental porcine-derived CM. Two regions of interest (ROIs) were defined and pictures were taken on EDD-10 (directly before treatment), EDD-11, and EDD-14 with an intravital fluorescence microscope to display the potential increase in vessel density over time (see Figure 3).

The analysis could show a clear increase of vessel density over time in both groups. Starting on EDD-10 at a density level of 0.047 µm/µm^2^ (95%-CI [0.0023–0.0072]) in the control group and of 0.035 µm/µm^2^ (95%-CI [0.0015–0.0553]) in the ESWT group, vessel density significantly increased within one day up to 0.0086 µm/µm^2^ (95%-CI [0.0054–0.0118]) in the untreated and 0.0075 µm/µm^2^ (95%-CI [0.0047–0.0104]) in the ESWT-treated group. Statistically significant differences between both groups could be detected neither on EDD-10 (*p* = 0.344) nor on EDD-11 (*p* = 0.630). Vessel density further increased up to EDD-14, with a mean density of 0.0160 µm/µm^2^ (95%-CI [0.0160–0.0188]) in the control and 0.0179 µm/µm^2^ (95%-CI [0.0152–0.0207]) in the ESWT group (*p* = 0.279) (see Figure 4a). Here, specimens treated with ESWT showed a stronger increase of vessel density between EDD-11 and EDD-14 (+ESWT: 241.25%, 95%-CI [44.41–438.01] vs. −ESWT: 165.39%, 95%-CI [−21.62–352.41]), although this difference could not be shown to be statistically significant (*p* = 0.441) (see Figure 4b).

#### 2.1.3. Vessel Junction Analysis

As branching of blood vessels is a critical process in angiogenesis at sites where new vessels are formed from pre-existing vessels, a vessel junction analysis was performed to further investigate the effects of ESWT on this parameter (see Figure 3).

The statistical analysis showed a significant increase in the number of vessel junctions over time for both groups. Before treatment, the number of junctions per area was 1.90 (95%-CI [0.706–3.094]) within the control and 1.64 (95%-CI [0.0289–2.997]) within the ESWT group and did not differ significantly (*p* = 0.740) between the groups. One day after treatment, the number of junctions increased up to 8.25 (95%-CI [4.354–12.146]) junctions per area in the sham-treated and 5.89 (95%-CI [2.970–8.808]) in the ESWT-treated group (*p* = 0.268). While the analysis could not show a statistically significant difference on EDD-11, the number of junctions at EDD-14 was considerably higher after ESWT treatment (+ESWT: 26.56, 95%-CI [21.792–31.319] vs. −ESWT: 21.19, 95%-CI [14.315–28.060]; *p* = 0.146) (see Figure 5a). This could be confirmed by calculating the increase in vessel junctions over time between EDD-11 and EDD-14. After ESWT-treatment, the number of junctions per area did show a statistically significant higher increase (751.77%, 95%-CI [110.54–1393.00]) compared to the control group (363.23%, 95%-CI [−50.3–776.75]; *p* = 0.068) (see Figure 5b). These findings support the assumption of a pro-angiogenetic effect of ESWT during the ingrowth of a porcine-derived CM.

### 2.2. Gene Expression Analysis

Quantitative real-time PCRs were performed to analyze the potential modulation of angiogenesis and immunoreaction by ESWT during ingrowth of porcine-derived CM. Here, the gene expression of HIF-1α, VEGF, MMP-9, MMP-13, and iNOS2 were measured.

#### 2.2.1. HIF-1α

HIF-1α was measured at EDD-11 and EDD-14 (one- and four-days post-treatment) to analyze the short- and long-term effect of ESWT on gene expression. Twenty-four hours after shock-wave therapy, HIF-1α-gene expression was considerably enhanced (2.03, 95%-CI [−2.117–6.168]) compared to the control group (0.65, 95%-CI [0.562–0.742]; *p* = 0.684), whereas mRNA levels assimilated towards EDD-14 (+ESWT: 1.06, 95%-CI [−0.667–2.793] vs. −ESWT: 0.58, 95%-CI [0.200–0.968]; *p* = 0.627). Although, on neither measuring point, a statistically significant difference could be detected between the groups (see Figure 6), slightly higher gene expression values 24 h after ESWT could indicate a potential pro-angiogenetic effect through HIF-1α-gene induction.

#### 2.2.2. VEGF

VEGF mRNA was measured at two time-points: one and four days post-treatment, to monitor differences in VEGF gene expression over time. Although the statistical analysis could not show a significant difference, neither on EDD-11 (*p* = 0.570) nor on EDD-14 (*p* = 0.459), VEGF-levels were slightly higher in the ESWT group (2.44, 95%-CI [−0.494–5.372]) compared to the control group (1.42, 95%-CI [0.324–2.511]) directly after therapy and decreased over time. No considerable difference could be detected after a latency of four days between both groups (+ESWT: 1.54, 95%-CI [1.303–1.782] vs. −ESWT: 1.09, 95%-CI [−0.011–2.192]) (see Figure 7). These results might indicate a direct VEGF-inducing effect of ESWT explaining the pro-angiogenetic properties shown within the microscopic analysis.

#### 2.2.3. MMP-9

MMP-9 mRNA levels were analyzed at EDD-11 and EDD-14. One day after treatment, significantly higher gene expression values could be detected within the ESWT group (1.08, 95%-CI [0.147–2.010]) compared to the control group (0.29, 95%-CI [−0.03–0.602]; *p* = 0.058). Although the measurements showed considerably higher MMP-9 levels at EDD-14 within the ESWT-treated group (+ESWT: 4.62, 95%-CI [0.759–8.473] vs. −ESWT: 3.37, 95%-CI [1.029–5.704]), the analysis did not show a statistical significance (*p* = 0.232) (see Figure 8). MMP-9 is known to be a stimulator of angiogenesis, endothelial cell migration, sprouting, and neovascularization. It is secreted by heterophil cells of the avian immune system, thus these results indicate a direct immunomodulating effect by ESWT potentially enhancing the tissue’s wound healing capacity and angiogenesis.

#### 2.2.4. MMP-13

MMP-13-gene expression levels were measured one and four days after treatment to analyze the short- and long-term effect of ESWT on the local immune system. Even though slightly higher MMP-13 levels could be detected at EDD-11 within the control group (−ESWT: 40.41, 95%-CI [20.638-60.178] vs. +ESWT: 29.27, 95%-CI [9.939–48.596]), no statistically significant difference could be shown (*p* = 0.725). Within the following three days, MMP-13-gene expression levels dropped considerably. Here, the ESWT group (1.21, 95%-CI [−1.27–3.699]) showed significantly lower expression values compared to the control group (5.78, 95%-CI [0.755–10.811]; *p* = 0.01) (see Figure 9).

#### 2.2.5. iNOS2

The measurements showed slightly higher iNOS2 gene expression values within the control group (40.78, 95%-CI [20.652–60.900]) compared to the ESWT group (22.61, 95%-CI [4.464–40.76]) with no statistical significance (*p* = 0.135) on EDD-11. Gene levels dropped considerably over the following three days still showing higher values within the sham-treated specimens (−ESWT: 1.84, 95%-CI [0.116–3.564] vs. +ESWT: 0.54, 95%-CI [−0.795–1.872]; *p* = 0.131) (see Figure 10). These results indicate an immune-modulating effect of ESWT-treatments, potentially influencing macrophage-activity.

### 2.3. Immunohistochemical Staining of CM Integration into CAM Tissue

To evaluate the integration of the CM into the CAM tissue, paraffin-embedded tissue sections were prepared for exemplary immunohistochemical staining (see Figure 11). Hematoxylin and eosin (HE) stainings served for evaluation of the position of the transplant within the CAM tissue and staining against α-SMA was used to detect vessels within the tissue.

It was clearly seen that CM were integrated into the CAM tissue and in parts were fully surrounded by CAM tissue bearing a high number of blood vessels. At the borders of the transplant, the invasion of cells from the CAM tissue into the porous structure of the CM could be seen. A slight tendency could be detected that within samples that received ESWT, the invasion of cells from the surrounding CAM tissue was more pronounced compared to CM that were sham-treated. Furthermore, in CM treated with ESWT, blood vessels were visible in the immediate vicinity of the transplant. Due to degradation of the porous transplant it was difficult to evaluate whether blood vessels fully invaded the transplant; however, a well vascularized tissue surrounding the whole transplant was detected, pointing to its successful integration.

## 3. Discussion

Dental CM are used whenever autologous tissue is not available in sufficient quantity and/or quality. CM are the standard material for the therapy of aesthetically and functionally disturbing recessions, the enlargement of the attached gingiva, but also the closure of extraction sockets and the thickening of the peri-implant soft tissue in the oral cavity. Since the removal of autologous grafts results in increased local morbidity, CM offer a safe alternative. However, a strong initial or a prolonged inflammatory response often leads to impaired healing and integration. This potentially results in inflammation, scarring, or transplant rejection. Thus, the focus of this study was to combine a frequently used allogeneic material for soft tissue regeneration with ESWT, a combination that was not previously assessed in the literature.

There is an increasing number of implantable medical devices and materials, which are continuously modified to induce a beneficial tissue remodeling and healing. However, in the end the procedures’ success still relies to a certain degree on the body’s own self-healing capability. Strengthening of the individual ability to integrate the implanted foreign material may serve as a new approach to guide the body’s endogenous regenerative capacity to achieve a favorable tissue repair. ESWT is an FDA-approved therapy for musculoskeletal disorders and more and more it is also suggested for several other areas of application [15]. ESWT is said to improve the tissue vascularization and to help to modulate the immunological response to foreign materials [21,23,26,34]. It serves as a new approach to guide the body’s endogenous regenerative capacity to achieve a favorable tissue repair. This hypothesis was evaluated in the present analysis.

The present analysis of the vascularized area of the collagen matrix could confirm the reported beneficial effects of ESWT on angiogenesis showing a significantly higher proportion of vascularized area per CM four days after treatment compared to the control group. Furthermore, the increase in vascularization within three days was significantly higher within the ESWT treated group, supporting the impression of a pro-angiogenetic effect of ESWT in the medium term. Even though no statistically significant effect could be detected, the vessel density analysis did show slightly higher values at EDD-14 after ESWT and a higher increase of vessel density between EDD-11 and EDD-14. Additionally, the vessel diameter observed in the ESWT treated group tended to be larger compared to the control group (see Figure 1). The vessel junction analysis could show a similar trend towards a pro-angiogenetic effect of the ESWT. Even though lower values were observed after twenty-four hours in the treatment group, the increase in the number of vessel junctions from EDD-11 to EDD-14 was more pronounced after ESWT compared to the controls.

As branching of blood vessels is a process occurring during neo-angiogenesis at sites where new vessels are formed from pre-existing vessels, the vessel junction analysis did further confirm the impression of a richer vascularization of the transplant after ESWT treatment. The successful integration and high peri-transplant vascularization was confirmed via histological analysis showing the CM to be fully surrounded by CAM tissue bearing a high number of blood vessels.

These observations are consistent with the changes after ESWT described in literature so far. The increase in blood perfusion and the abundance of microvessels observed in rodent models upon ESWT could be related to the induction of VEGF and PCNA expression through ESWT [21,23,34].

The lower vascularization parameters observed one day after the treatment could result from variances within the analyzed population. However, they could also stem from a short-term stagnation of vessel growth immediately after activation by the ESWT. It is possible that activation following ESWT is the result of a direct mild harming effect of ESWT leading to cell decay and formation of reactive oxygen species [35], followed by a strong, compensating pro-angiogenetic impulse. Treating human foreskin fibroblasts with 100 impulses at 0.19 mJ/mm^2^, Basoli et al. observed a short-term inflammatory response through reactive oxygen species, interleukin production and monocyte chemoattractant protein 1 followed by cell proliferation suggesting a protecting response after an initial state of low-level inflammation [35].

Negative effects of ESWT on cell viability have already been observed. Mattyasovszky et al. showed a lower cell viability in human skeletal muscle cells after application of higher energy flux density (EFD) values (>0.14 mJ/mm^2^) [36], whereas EFDs greater than 0.31 mJ/mm^2^ appeared to be cytoreductive [37]. For the present study, an EFD below this potentially harmful limit was selected (0.12 mJ/mm^2^). The number of impulses was adjusted according to studies comparing the effect of different impulse rates on cell proliferation, tissue remodeling, and inflammatory activity [38,39]. Up until now, only a few studies investigated whether multiple sessions of ESW application can improve the therapeutic outcome; the data is inconsistent. De Lima Morais et al. could show a higher skin thickness after two ESWT applications, whereas the single application showed lower values [18]. The same applies to neo-collagenesis and neo-angiogenesis as well as related gene expression. On the other hand, Kuo et al. showed that an increase in the number of applications could not contribute to a reduction of the necrotic zone in a skin flap model [34].

The embryogenic development is a sensitive stage. Kiessling et al. reported a dose-dependent increase in deaths of three- and four-day-old chicken embryos after ESWT; surviving embryos did show severe congenital defects [40]. A considerable high vulnerability is present during organ development. Therefore, ESWT was performed in a later stage of development in the present study. Preliminary studies of our working group could not show any signs of congenital malformation or a higher rate of deaths after exposure to 100, 200, and 500 impulses of 0.12 mJ/mm2 ESWT at day 11. Considering that early ESWT application may contribute to embryonic harm and that the treatment and observation period ranges from 3–14 days, a single ESWT application was chosen for this study. In further studies, the effects of multiple applications as well as the influence of different EFDs and frequencies will be investigated.

Gene expression analysis via qPCR was performed to assess the impact of ESWT on different pro-angiogenic factors. Twenty-four hours after ESWT, gene expression of the transcriptional factor HIF-1α, which is known for its pro-angiogenetic and cell proliferation promoting effects [41] was considerably higher compared to sham-treatment after twenty-four hours. Within the following three days, HIF-1α expression decreased in both groups still showing higher values within the ESWT group. Additionally, VEGF, being the most potent endothelial-specific mitogen promoting angiogenesis [41], was analyzed, showing similar trends in the genes expression analysis supporting the impression of the pro-angiogenetic effect of ESWT on a molecular level.

The results must be seen in the context of the natural development of the CAM tissue. The CAM exhibits an intrinsically high mitotic rate until EDD-10; thereafter, the mitotic rate begins to decline. Thus, there is a natural increase in vascularization within the first ten days of development [42]. Within this timeframe, it is difficult to assess the impact of treatments that should stimulate the induction of angiogenesis, because there is naturally a huge amount of neovascularization. Therefore, for measurements of the angiogenic response, later timepoints from EDD-10 onwards were chosen deliberately to be able to assess neovascularization under more static conditions after the highly vasoproliferative phase of the CAM tissue. Thus, ESWT had to induce a vasoproliferative response in parallel with the naturally reduced rates of growth of the CAM’s endothelium. This might explain why no more prominent effects were seen.

In addition to its ability to stimulate neo-angiogenesis, ESWT is regarded as a possibility to modulate immune responses to foreign materials [21,23,26,34]. Of course, immune responses in adult mammals cannot be directly compared to the avian embryogenic system; however, macrophage-like cells were found shortly after the start of the egg development. Studies could also show their phagocytic activity which suggests homologies to mammalian fetal macrophages [33]. Between EDD-10 and EDD-14, two major cell types of the immune system are known in chicken embryos. These are on the one hand heterophils, which are regarded as the avian analogue to mammalian neutrophils. Heterophils are the main source of MMP-9 in the chicken embryos; thus, this protein can be regarded as a marker for these cells. The second embryonal type of immune cells are regarded as members of the monocyte/macrophage lineage. These cells are the major source of MMP-13 in the chicken embryo [43,44,45]. Matrix metalloproteinases (MMPs) are extracellular zinc-dependent endopeptidases that are involved in the degradation and remodeling of extracellular matrix in several physiological and also pathological processes. It is well known that different MMPs such as MMP-9 and MMP-13 play a major role in neovascularization. The sprouting of new blood vessels is an invasive process, which requires the proteolytic activities provided by MMPs [46,47,48,49]. In this context, heterophils and macrophages might therefore play a special role as suppliers of these MMPs to the sites of tissue remodeling and neovascularization around the transplant.

MMP-13 has already been associated with collagen remodeling and the onset of angiogenesis in the CAM assay. It was shown to be induced rapidly upon an angiogenic stimulus and its expression coincided with the early events in vessel formation [43]. As such, MMP-13 seems to be a factor that initiates angiogenesis. This idea is also supported by the finding that MMP-13 is able to promote VEGF secretion from endothelial cells and fibroblasts [48]. This might explain why a decrease in MMP-13 gene expression from EDD-10 to EDD-14 was observed. At EDD-10, blood vessel formation had already been initiated and first vessel sprouting around the transplant could be seen. Furthermore, the CAM tissue did already leave its highly vasoproliferative phase as described above. Possibly, earlier timepoints within the CAM development and timepoints immediately after insertion of the CM onto the CAM tissue would give higher MMP-13 gene expression values. Perhaps this protein is simply no longer needed for further ingrowth of the transplant at the later stages.

The development of the gene expression of MMP-9 was in the opposite direction compared to MMP-13. While MMP-9 mRNA levels were low on EDD-10, its level clearly increased until EDD-14; moreover, ESWT was able to slightly increase MMP-9 expression. Within the last years, more and more data has accumulated that shows the importance of the recruitment of cells of the immune system to sites of neovascularization. In this context, neutrophils were found to play an important role, especially by providing MMP-9 to the sites of vessel formation [49]. This effect seems to be relevant in tumor angiogenesis [50,51] as well as in cases of tissue transplantation [49], and biomaterial tissue integration [52]. MMP-9 is stored by neutrophils in secretory granules and released upon a corresponding stimulus as pro-MMP-9. Unique for neutrophils and chicken embryo heterophils is their ability to secrete TIMP-free pro-MMP-9—this means MMP-9 without its inhibitor—which is readily available for proteolytic activation [53]. It has been shown that the disruption of the influx of heterophils, and thus the delivery of MMP-9 to the sites of neovascularization, significantly inhibited angiogenesis [44]. This underlines the importance of immune cells for angiogenesis as well as for tissue remodeling, which is associated with any kind of wound healing. One mechanism of action that is proposed for the proangiogenic effects of MMP-9 is the degradation of the extracellular matrix by the protein that facilitates tissue remodeling and thereby also activates bound factors such as VEGF [49]. Neutrophils are considered to be matrix programmers which play an important role in any kind of tissue regeneration [52]. All the more interesting is that we observed that ESWT could be able to elevate the expression of the main neutrophil marker MMP-9. Furthermore, it is possible that the cells that were seen to invade the transplant in the immunohistochemical stainings (see Figure 11) were neutrophils, which could be subject to further investigations. Maybe they are able to guide the vessel sprouting into the CM and thus the effective tissue integration.

In mammalian cells, iNOS is a cell expression marker for M1 macrophages and serves as a marker for pro-inflammatory action within our model. Even though the exact immunological process is largely unexplored in avian species, similarities to murine and human models are assumed, especially regarding an enhanced iNOS activity after contact of macrophages with pathogens [33]. Although not statistically significant, the iNOS gene expression was observed to be higher within the sham-treated group compared to the ESWT group, which might imply a greater pro-inflammatory foreign-body response on the CM. The putative anti-inflammatory effects of ESWT were investigated by Ciampa et al., demonstrating an increase in NO production by enhancing the catalytic activity of neuronal NO-synthase, and downregulating of NF-κB and NF-κB dependent genes (e.g., iNOS and TNF-α) [38,54]. In rodent models, wound healing was enhanced after ESWT, showing a decrease in leukocyte infiltration and higher levels of VEGF and eNOS [26].

The high variances that were observed in the gene expression analysis seemed to be based on different aspects which could not all be fully resolved within the scope of this study. First of all, one has to bear in mind that the samples for the qPCRs were isolated from living animals not from cell lines cultivated in vitro, which results in a certain inter-individual variability. Secondly, we are dealing with different stages of embryonal development, which are quite delicate. A certain period of time was covered by our study, not only a single timepoint, which poses a special challenge for the qPCR. It is difficult to find a reference gene that is stable within the whole period of embryonic development, which is characterized by huge changes within the gene expression profile within only several days. Additionally, egg development is only synchronized by the start of breeding; thus, not all eggs develop exactly at the same pace, and differences between the eggs can be observed visually, which must be assumed to be also reflected in the gene expression profile. Indeed, the whole qPCR sample set was analyzed twice, and several aspects of the method could already be optimized. The whole data relies on three completely independent runs of the experiment, with each run containing all four experimental groups, represented by initially five to ten individual eggs per group. The sample size for the qPCR was unfortunately reduced by the drop-out rates.

Further analysis showed that the drop-out rate after ESWT was close to zero and showed no difference between the groups. The remaining dropouts occurred within earlier stages of embryo development. Thus, no correlation with ESWT can be assumed. The high overall rates of dropout can be divided into two groups: early and late losses. Early loss can be caused by a missing fertilization of the eggs, transport damage, developmental disorders, damage caused by the opening of the egg or the removal of the albumen, as well as infections by fungi/bacteria. Late losses, on the other hand, are less easily definable. An infection at a later stage of development can lead to the death of the embryo, as can developmental disruption or direct mechanical damage during treatment or sham-treatment. The observed dropout rates are not unusual for the CAM assay, as the opened egg is a rather fragile study object, and we have to keep in mind that we work with animals that are still in the embryonic development.

Unfortunately, the different genes posed different challenges to the qPCR because primers that function in the chicken transcriptome had to be found, gene expression of certain genes on certain days were near the detection limit, and in some samples, RNA isolation failed or certain melting curves revealed unspecific primer binding, and thus, were excluded from the analysis. A further point complicating the evaluation is the limited number of studies analyzing the avian embryogenic immune system. Details about the cells existing in the avian embryogenic system during different stages of development are unknown; additionally, not much is known about the cellular markers expressed by these cells. This costed a lot of establishing work, which will be intensified in the scope of further studies with this model system.

Apart from these disadvantages, the CAM assay also provides great benefits. It allows in vivo analysis of angiogenesis and immunoreaction on biomaterials within a living organism with a minimum of stress and suffering of the experimental animal and is considered to be the bridge between in vitro and conventional rodent animal models [31].

Regarding the number of studies showing clear signs of chronic inflammation and fibrosis along with foreign body giant cells surrounding the tissue [1,55,56,57], the research on possibilities to strengthen the individual ability to functionally and structurally integrate biomaterials becomes more and more important. Using the CAM assay, we could show enhanced angiogenesis and neovascularization in CM after ESWT. Furthermore, ESWT could induce a reduction of inflammatory activity after a latency of four days. These findings point to the potential value of ESWT to facilitate tissue integration of transplanted biomaterials and future studies are highly recommendable to further investigate this promising approach.

## 4. Materials and Methods

### 4.1. Collagen Matrices

Mucoderm^®^ (CM; botiss biomaterials GmbH, Zossen, Germany) is a porcine-derived, three-dimensional, native collagen matrix composed of collagen type I and type III without further chemical treatment. It is used as a soft tissue graft avoiding the need for autograft harvesting. Its porous collagen network is meant to act as guiding structure for cell migration. In oral and maxillofacial surgery, it is used for soft tissue augmentation, closing of extraction sockets, and thickening of peri-implant tissue [58].

### 4.2. In Ovo Chorioallantoic Membrane Assay (CAM Assay)

Fertilized white Leghorn chicken eggs (LSL Rhein-Main GmbH, Dieburg, Germany) were incubated at 38 °C with constant humidity in an incubator (Type 3000 digital and fully automatic, Siepmann GmbH, Herdecke, Germany). For the first three days, eggs were placed horizontally on one side to ensure the CAM would detach from the upwards pointing eggshell. On EDD-3, eggs were prepared by removing 5–6 mL of the albumin in order to enlarge the space between eggshell and CAM. A small window of 3 × 2 cm was cut into the upwards pointing part of the eggshell (Figure 12a). The window was covered with Parafilm^®^ (Sigma-Aldrich, St. Louis, MO, USA) to prevent evaporation. On EDD-7, CM were cut into pieces of 25 mm^2^ and placed onto the CAM (in total, 28 eggs per group within three independent runs) (see Figure 12b). On day 10 of egg development, ESWT was applied once in the treatment group as described in Section 4.3, After 24 h, pictures were taken with a digital intravital fluorescence microscope (Olympus BXFM, Olympus Deutschland GmbH, Hamburg, Germany) at a 100-fold magnification using the cellSens Dimension software package. A first group of embryos (in total nine eggs per group within three independent runs) was euthanized on EDD-11 to assess the immediate tissue response to ESWT by mRNA analysis via qPCR. The remaining embryos were further incubated, and vascularization was analyzed again four days after ESWT with the digital intravital fluorescence microscope. Then, they were euthanized on day 14 by decapitation and a sample of the CM and the surrounding CAM was extracted for qPCR or immunohistochemical staining (see Table 1).

### 4.3. Extracorporeal Shockwave Therapy (ESWT)

For application of ESWT, eggs were embedded in an open egg carton. The cavity was filled with 0.9% NaCl solution (B. Braun Melsungen AG, Melsungen, Germany) and the radial handpiece of the Swiss DolorClast Classic (E.M.S. Electro Medical Systems, Nyon, Switzerland) was set in a laboratory stand. The tip of the 15 mm applicator was immersed by slowly lowering it into the solution above the CM (see Figure 13). The air pressure was set to 3.0 bar with a frequency of 5 Hz at a distance of 5 mm to the applicator corresponding to an EFD of 0.12 mJ/mm^2^. A total of 500 impulses were applied. After application, the handpiece was raised from the window and cleaned to avoid contamination. NaCl was removed from the egg and the window was covered with Parafilm^®^ for further incubation. The control groups were sham-treated by filling the cavity with 0.9 % NaCl solution and removing it after an equivalent period of time.

### 4.4. Intravital Fluorescence Microscopy

Intravital fluorescence microscopy was performed using a in vivo microscope (Olympus BXFM, Olympus Deutschland GmbH, Hamburg, Germany) at a 100-fold magnification and the cellSens Dimension software package by Olympus.

#### 4.4.1. Determination of the Vascularized Area on the Collagen Matrix

The analysis was performed in all groups at EDD-11 and EDD-14 of egg development. Using the automated composing-function of the software package, multiple image arrays were generated showing the complete CM and smaller parts of the surrounding soft tissue. After picture acquirement, the data were pseudonymized. The measurements were done by the same examiner to avoid interobserver-variability. First, the outer border of the transplant was traced, then the inner border of the vascularized area was marked. The software automatically calculated the marked area. To calculate the vascularized area, the result of the first measurement was subtracted from that of the second measurement (see Figure 14). The proportion of the vascularized area per CM as well as the increase of the vascularized area between EDD-11 and EDD-14 were calculated.

#### 4.4.2. Vessel Density and Vessel Junction Analysis

Using intravital fluorescence microscopy, the focus was set at two pre-determined regions of interest (each one in a corner of the CM) and pictures were taken at EDD-10 (directly before treatment), EDD-11 and EDD-14. Afterwards, the data were pseudonymized. The analysis of the vessel density was carried out by one examiner to avoid interobserver-variability. Using the software package cellSens Dimension, a grid of 1000 × 1000 µm was laid over the image. The vessels visible inside of the ROI were marked and the software calculated the length of all vessels (vl) marked. The overall length of all vessels (VL) was computed by summation and the vessel density (vd) per µm^2^ was calculated by dividing the overall vessel length by the marked area of 1,000,000 µm^2^. As two areas per CM were analyzed, the received values were summed (VD).

Underlying equation:vl_1_ + vl_2_ + … + vl_x_ (µm) = VL (µm)VL/1,000,000 µm^2^ = vd (µm/µm^2^)vd1 + vd_2_ = VD (µm/µm^2^)(1)

Furthermore, the increase of vessel density between EDD-11 and EDD-14 was calculated.

A vessel junction analysis was performed using the acquired data saved in JPEG image format. The images were edited using the open source software Fiji, a distribution of ImageJ (see Figure 15d). An automated analysis calculating the number of branches and junctions per image was performed followed by manual re-counting (interobserver agreement: 100%). The absolute number of vessel junctions (vj) per ROI was summarized for both ROIs per CM.

Underlying equation:(vj_1_ + vj_2_)/2 = VJ(2)

Furthermore, the increase of vessel junctions per area between EDD-11 and EDD-14 was calculated.

#### 4.5. mRNA Isolation and Quantitative Polymerase Chain Reaction (qPCR)

At EDD-11 and -14, in order to isolate the total cellular mRNA, CAM and CM samples were cut out, immediately transferred to cryotubes, and shock-frozen in liquid nitrogen. Digestion of the tissue was performed with the homogenizer Speedmill Plus (Analytik Jena AG, Jena, Germany) using innuSPEED Lysis Tubes E (Analytik Jena AG, Jena, Germany) according to manufacturer’s specifications. The following RNA isolation was performed with the RNeasy Mini Kit (Qiagen N.V., Venlo, Netherlands) according to manufacturer’s specifications including a DNase digestion. Concentration and purity of the obtained RNA were determined with the NanoDrop™ One (Waltham, MA, USA). Transcription of RNA to cDNA was performed with the iScript^TM^ cDNA Synthesis Kit 100 (Bio-Rad Laboratories Inc., Hercules, CA, USA) according to the manufacturer’s specifications. Obtained cDNAs were stored at −20 °C. SYBR green based qPCRs were performed in duplicate, including 5 µL cDNA in each PCR reaction using the iTaq Universal SYBR Green Supermix (Bio-Rad Laboratories Inc., Hercules, CA, USA) with the CFX Connect Real-Time PCR Detection System (Bio-Rad Laboratories Inc., Hercules, CA, USA). The beta actin gene was used as endogenous reference. Its suitability for normalization was determined with the Normfinder Excel Add-in (https://moma.dk/normfinder-software). The quantification of the transcripts was performed by the ∆∆CT method making use of an internal calibrator sample which was included in all qPCR runs. Gene expression levels are relative levels compared to the internal calibrator. qPCR analyses were performed for HIF-1α, VEGF, and MMP-9 and -13 as well as iNOS2 using the primers listed in Table 2. Data analysis was performed with the Bio-Rad CFX Manager software (Bio-Rad Laboratories Inc., Hercules, CA, USA).

### 4.6. Immunohistochemical Staining

At EDD-14, for preparation of samples for immunohistochemical staining, the embryo was sacrificed by decapitation. The CAM bearing the CM was then excised with sterilized surgical scissors and placed onto a filter paper stripe. The sample was transferred into a plastic cassette (Carl Roth GmbH + Co. KG, Karlsruhe, Germany), immobilized, and put into a formalin solution (4%) (VWR International bvba, Leuven, Belgium) for 24 to 72 h. Afterwards, the plastic cassette was removed from the 4% formalin solution, washed three times with purified water for 20 min each, and incubated in isopropanol solution with increasing concentrations (80%/90%/100%) for one hour in each solution. The cassette was then washed with purified water and incubated in xylene (AppliChem GmbH, Darmstadt, Germany) for 24 h. Each specimen was embedded in paraffin and cut into 3 µm slides with a microtome (Leica CM1900, Leica Biosystems Nussloch GmbH, Nußloch, Germany).

For HE stainings, paraffin was removed from the slide and the specimen was incubated in Mayer’s hemalum solution (Carl Roth GmbH + Co. KG, Karlsruhe, Germany) for 5 min. Subsequently, each slide was again washed in purified water and incubated in eosin (Merck KGaA, Darmstadt, Germany) for 2 min. Slides were then incubated in isopropanol solution with increasing concentrations (80%/90%/100%) for 2 min in each solution and then in xylene for 10 min. Finally, slides were prepared for microscopy by embedding the specimen in Eukitt^®^ (Sigma-Aldrich, St. Louis, MO, USA).

Immunohistochemical staining for Alpha Smooth Muscle Actin (α-SMA) allowed visualization of vessels within the CAM. Slides were dewaxed with xylene and isopropanol solutions with decreasing concentrations (100%/90%/80%/70%) for 5 min in each solution and cooked in citrate buffer (pH 6.0) for 30 min. After preparation slides were incubated with monoclonal α-SMA antibodies (Sigma-Aldrich, St. Louis, MO, USA) dissolved in (1/1500) phosphate-buffered-saline (PBS) + 1% bovine serum albumin (BSA). As secondary antibody biotinylated polyclonal goat anti-mouse immunoglobulin (Dako Denmark A/S, Glostrup, Denmark) was added and visualized using horseradish peroxidase-conjugated Streptavidin (1/250) (Dako Denmark A/S, Glostrup, Denmark). Finally, slides were again prepared for microscopy by embedding the specimen in Eukitt^®^ (Sigma-Aldrich, St. Louis, MO, USA).

Microscopical slides were analyzed using the Nikon Eclipse TE2000 Inverted Microscope (Nikon Corp. Chiyoda, Japan) and digitalized by transferring the image from the inbuilt camera system (Nikon’s DS-Fi3, Nikon Corp. Chiyoda, Japan) to Nikon’s analysis software NIS-Elements (Nikon Corp. Chiyoda, Japan).

### 4.7. Statistics

To analyze the differences between the measured values, normality (Shapiro-Wilk) and homogeneity of variance tests (Levene Statistic) were performed at first in order to check the conditions for the subsequent analysis. *p*-values were obtained with an Independent Samples T-Test. In case of not normally distributed values, a Mann-Withney test was used instead. The statistical analyses were performed using SPSS version 24 for Windows (IBM, Armonk, NY, USA); graphics were generated using Microsoft^®^ Excel Version 16.40 for Macintosh (Microsoft Corporation, Washington, DC, USA). A *p*-value ≤ 0.1 was termed significant with * *p* < 0.1 ** *p* < 0.05, *** *p* < 0.005. Values are displayed as mean and 95%-CI.

## Figures and Tables

**Figure 1 ijms-21-07574-f001:**
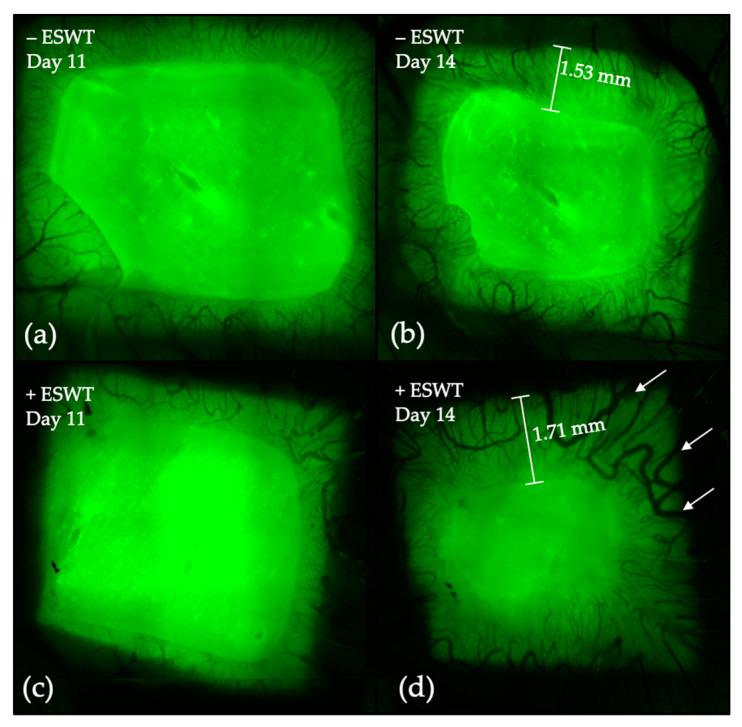
Exemplary multiple image arrays of treated and untreated specimens. (**a**) Intravital fluorescence image of the CM at EDD-11 in an egg of the control group. (**b**) Development of the vasculature up until EDD-14 in the control group. Vessels were growing into the transplant, forming a compact capillary network. (**c**) Twenty-four hours after therapy, the CM treated with ESWT showed a similar vascular density and vessel size compared to sham-treated one. (**d**) Within four days, the vascular network significantly increased. Compared to the sham-treated CM, the ESWT-treated specimen showed more and larger vessels (see arrows in (**d**)) as well as a richer capillary network. The vasculature covered a greater proportion of the transplant. Abbreviations: CM—collagen matrix; ESWT—Extracorporeal shockwave therapy.

**Figure 2 ijms-21-07574-f002:**
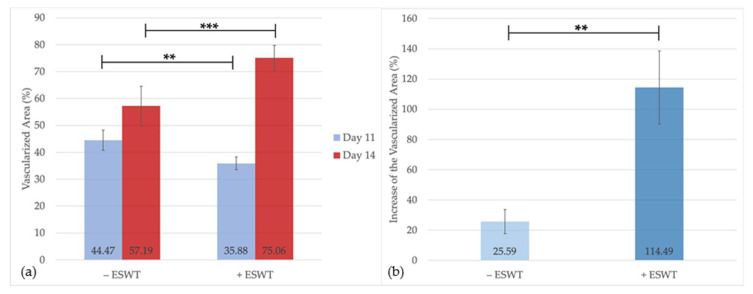
Results of the multiple image array analysis to assess the vascularized area of the CM (**a**) The graphs display the area percentage of the CM that was covered by vasculature (%). On EDD-11, the vascularized area of untreated transplants was shown to be significantly larger compared to that of the ESWT group. After a latency of four days post-treatment, the ESWT group showed a higher area percentage of vascularization than the control group. (**b**) The graph displays the increase of the vascularized area of the CM between EDD-11 and EDD-14 in the treated and the untreated group (%). After ESWT treatment, CM were shown to exhibit a significantly higher increase in the vascularized area compared to the control group. Shown are means ± SD, ** *p* < 0.05, *** *p* < 0.01, Independent-Samples *t*-Test between experimental groups and untreated control, normally distributed, homogeneity of variance.

**Figure 3 ijms-21-07574-f003:**
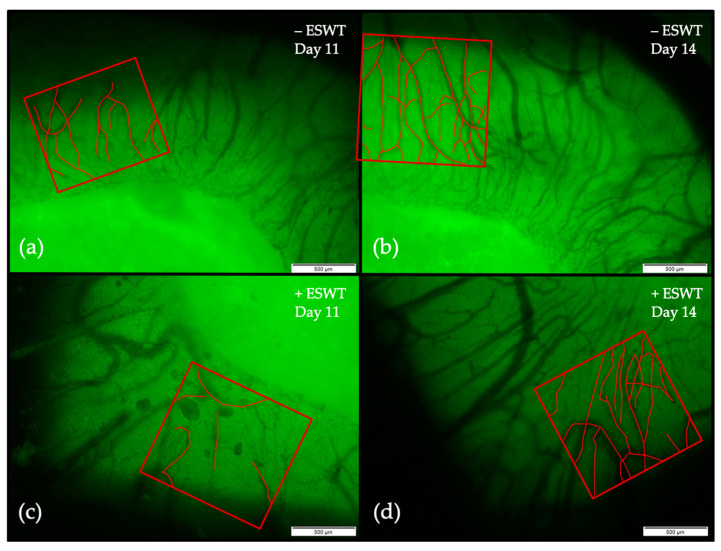
Exemplary pictures of the vessel density analysis. Due to autofluorescence, the brighter structure of the CM was well distinguishable from the surrounding soft tissue. In both groups, the vessel density increased between EDD-11 and EDD-14. (**a**) At EDD-11, the control group showed small and medium sized vessels growing into the transplant. (**b**) Three days later, the density of the vascular network was higher, vessels showed a larger diameter, and the area covered by the tissue was greater. (**c**) Twenty-four hours after ESWT, the vascular network was comparable to the one observed in the untreated specimen. (**d**) Within three days, the vessel density had significantly increased with more vessels of greater diameter growing into the CM compared to the control group. Note, the pictures of the vessel density analysis were further proceeded for vessel junction analysis according to description in the materials and methods section. The scale bar corresponds to a length of 500 µm.

**Figure 4 ijms-21-07574-f004:**
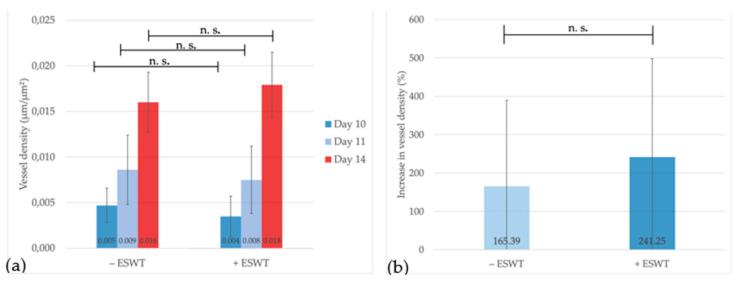
Vessel density analysis and comparison between treated and untreated groups within regions of interest (ROIs) over time. (**a**) The vessel density measured in different ROIs could be shown to increase between EDD-10 and EDD-14. No statistically significant difference could be shown between the ESWT-treated and the control group. (**b**) No significant difference could be shown regarding the increase of vessel density between EDD-11 and EDD-14, although there was a slightly greater increase within the ESWT group. Shown are means ± SD, n.s. = not significant. (**a**) Independent-Samples *t*-Test between experimental groups and untreated control, normally distributed, homogeneity of variance for EDD-10 and EDD-14; Mann-Whitney test, not normally distributed for EDD-11. (**b**) Mann-Whitney test, not normally distributed.

**Figure 5 ijms-21-07574-f005:**
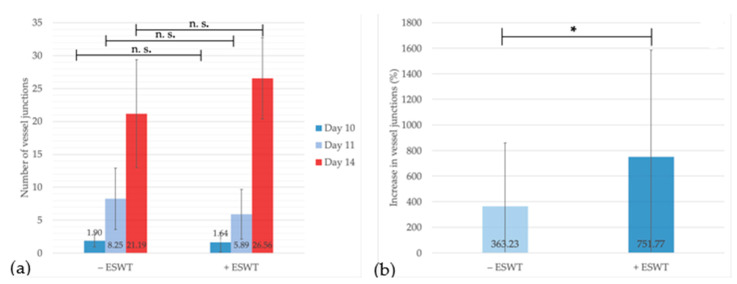
Vessel junction analysis within regions of interest over time. (**a**) Here, the number of vessel junctions between EDD-10 and EDD-14 is displayed. The number of junctions in treated and untreated groups was comparable until EDD-11. Four days post-treatment, the number of vessel junctions was slightly higher within the ESWT group compared to the control group. (**b**) The overall increase of junctions between EDD-11 and EDD-14 was significantly higher after ESWT. Shown are means ± SD, n.s. = not significant, * *p* < 0.1. (**a**) Independent-Samples T-Test between experimental groups and untreated control, normally distributed, homogeneity of variance. (**b**) Mann-Whitney test, not normally distributed.

**Figure 6 ijms-21-07574-f006:**
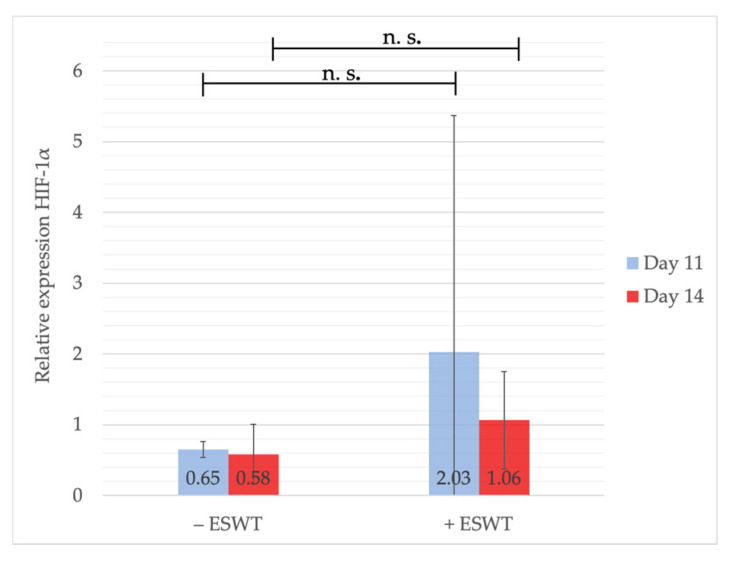
Gene expression analysis of HIF-1α via qPCR. Measurements were performed on EDD-11 and EDD-14, i.e., one and four days after ESWT or sham treatment. Although there was no statistically significant difference detectable, the analysis showed a slight increase of HIF-1α expression one day after ESWT compared to the control group, whereas no difference could be shown on EDD-14. Shown are means ± SD, n.s. = not significant. EDD-11: Independent-Samples T-Test between experimental groups and untreated control, normally distributed, homogeneity of variance. EDD-14: Mann-Whitney test, not normally distributed. Abbreviations: qPCR—quantitative polymerase chain reaction; HIF-1α—Hypoxia-inducible factor α.

**Figure 7 ijms-21-07574-f007:**
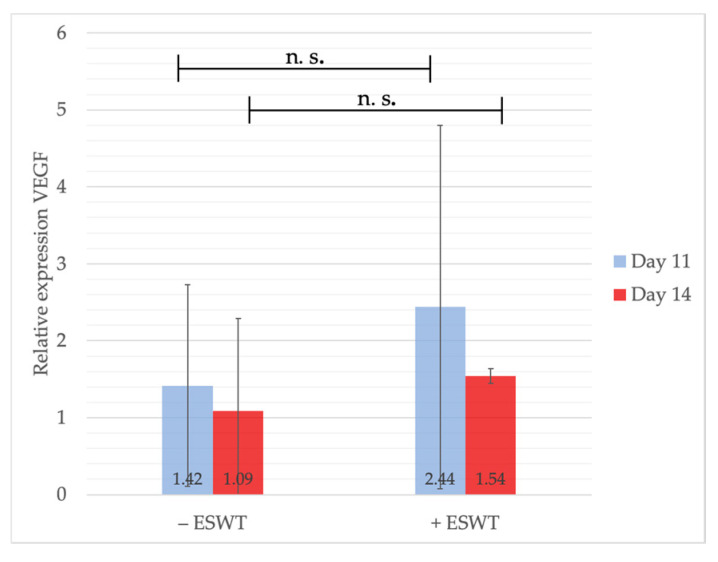
Gene expression analysis of VEGF via qPCR. The analysis could show a non-significant increase of VEGF one day after ESWT compared to the control group. At EDD-14 no effect of ESWT regarding VEGF expression was detectable. Shown are means ± SD, n.s. = not significant. Mann-Whitney test, not normally distributed.

**Figure 8 ijms-21-07574-f008:**
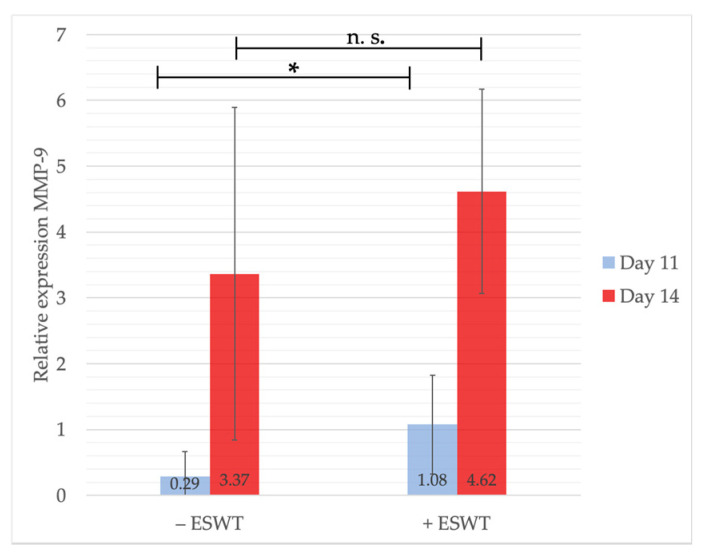
Gene expression analysis of MMP-9 via qPCR. The measurement on EDD-11 could show significantly higher levels in MMP-9 expression within the treated compared to the untreated group. Four days after ESWT or sham-treatment, MMP-9 was still higher within the ESWT group, although no statistically significant effect could be detected. Shown are means ± SD, n.s. = not significant, * *p* < 0.1, EDD-11: Mann-Whitney test, not normally distributed. EDD-14: Independent-Samples T-Test between experimental groups and untreated control, normally distributed, homogeneity of variance.

**Figure 9 ijms-21-07574-f009:**
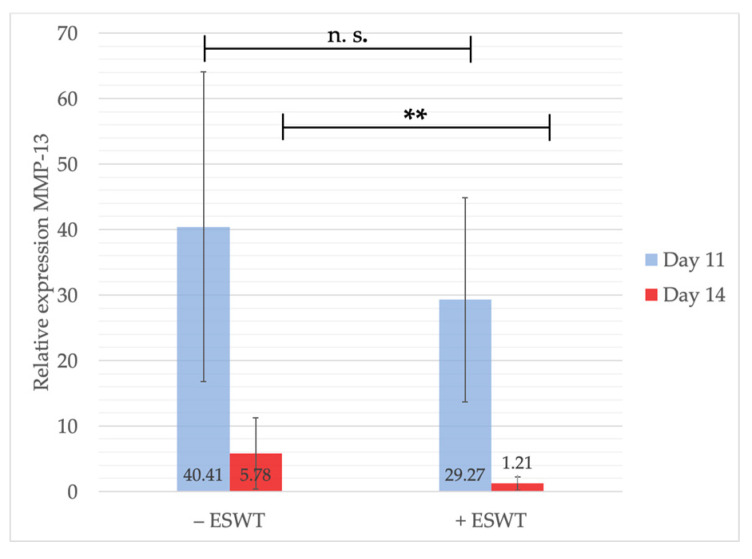
Gene expression analysis of MMP-13 via qPCR. Measurements were performed one and four days after ESWT-/sham-treatment. Slightly higher gene expression values could be detected within the control group on EDD-11. On EDD-14 statistically significant differences between treated and untreated groups were shown with a higher expression within the control group. Shown are means ± SD, n.s. = not significant, ** *p* < 0.05, EDD-11: Independent-Samples *t*-Test between experimental groups and untreated control, normally distributed, homogeneity of variance. EDD-14: inhomogeneity of variance.

**Figure 10 ijms-21-07574-f010:**
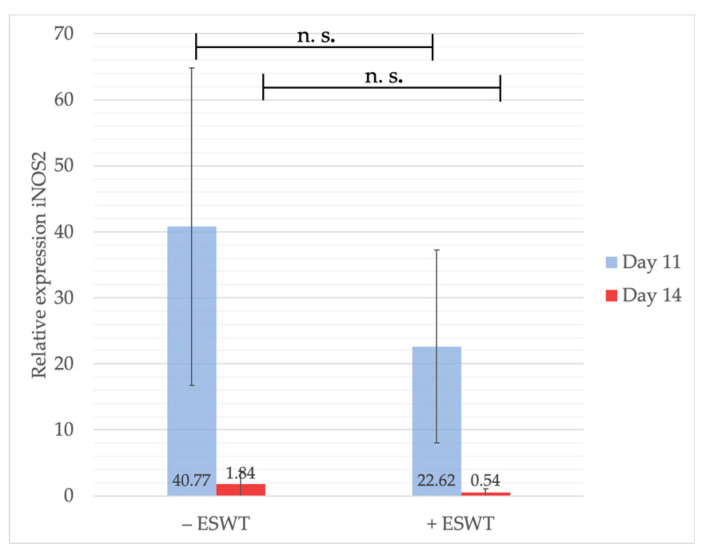
Gene expression analysis of iNOS2 via qPCR. Higher levels of iNOS2 could be shown within the control group compared to specimens treated with ESWT at EDD-11, although no statistically significant effect was detected. Four days after treatment, iNOS2 expression was still higher within the untreated control group compared to the ESWT-treated group. Shown are means ± SD, n.s. = not significant. EDD-11: Independent-Samples *t*-Test between experimental groups and untreated control, normally distributed, homogeneity of variance. EDD-14: inhomogeneity of variance.

**Figure 11 ijms-21-07574-f011:**
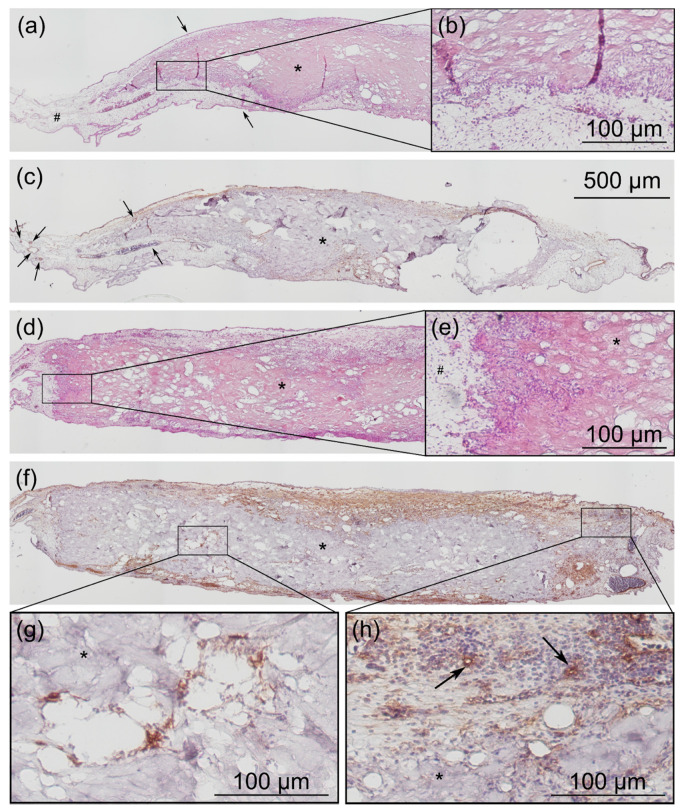
Immunohistochemical staining of CM integrated into CAM tissue. (**a**–**c**) show exemplary pictures of a transplant, which was sham-treated, (**d**–**h**) show a transplant that received ESWT. In (**a**) and (**d**), HE stainings are shown, in (**b**,**f**), we see staining against α-SMA of the same tissue section. (**b**,**e**,**g**,**h**) show the indicated tissue sections in higher magnification. # = chorioallantoic membrane tissue (CAM) * = CM transplant. (**a**) shows that the complete CM is surrounded by CAM tissue as indicated by the arrows. In the peripheral areas of the transplant, cells were invading into the porous structure of the CM, which can be seen in (**b**). Staining against α-SMA (**c**) shows several smaller and larger blood vessels in the CAM tissue, also in vicinity to the transplant. Larger blood vessels can be easily detected by the enclosed erythrocytes, which possess a nucleus in chicken. HE stainings of the ESWT treated CM (**d**) shows that there is the tendency that chicken cells invaded more deeply into the transplant at the margins (**e**) compared to sham-treated CM. Furthermore, cells were also found in the middle of the implant (**g**). Moreover, small capillaries were detected at the margins of the transplant (**h**).

**Figure 12 ijms-21-07574-f012:**
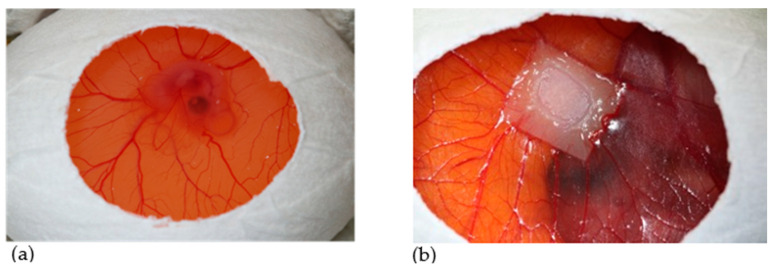
Experimental set-up. (**a**) Seven-day-old chicken embryo. After cutting a small window into the upper part of the eggshell, the window was closed with Parafilm^®^, and the eggs were further incubated. The in ovo CAM assay allows direct access to the extraembryonic chorioallantoic membrane via the open window. (**b**) Through the window, the ingrowth of soft tissue and vessels into the porcine-derived CM can be monitored.

**Figure 13 ijms-21-07574-f013:**
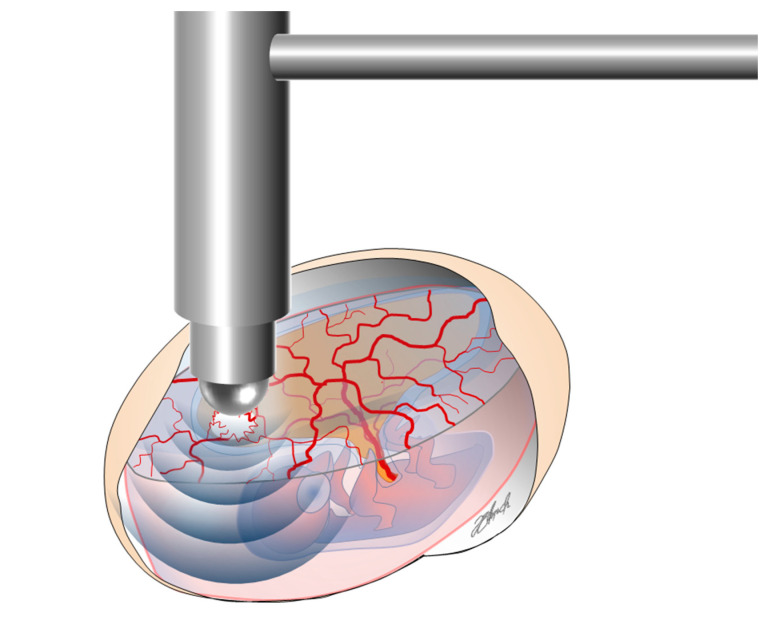
Experimental set-up for shock-wave application. The cavity was filled with 0.9% NaCl solution and the tip of the applicator was immersed into the solution. At an energy flux density of 0.12 mJ/mm^2^, 500 impulses were applied with the applicator centered above the CM.

**Figure 14 ijms-21-07574-f014:**
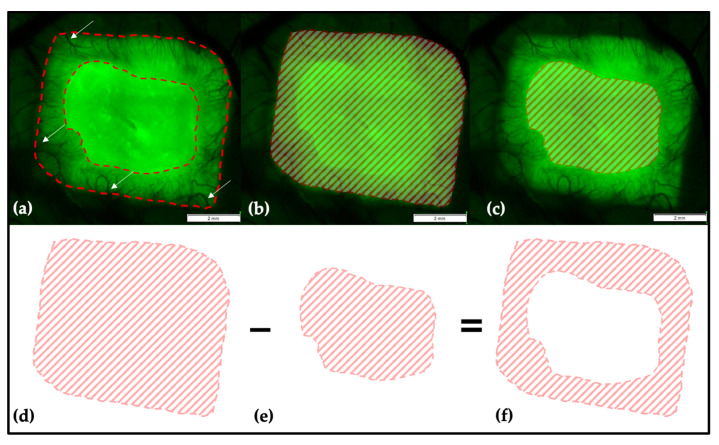
Multiple image array analysis of the CM. (**a**) The CM is visible as a brighter prismatic structure. Here, vessels can be seen growing into to transplant (arrows). The inner border of the vascularized area is marked and clearly distinguishable from the non-vascularized matrix. (**b**) To calculate the portion of vascularized area of the whole transplant, the outer border of the CM was traced, and the software package automatically calculated the encircled area as shown. (**c**) Then, the inner border of the vascularized area was marked. The software here also calculated the encircled area. To receive the size of the vascularized area (**f**), the non-vascularized area (**e**) in the center of the CM was subtracted from the area of the whole CM (**d**). The scale bar corresponds to a length of 2 mm.

**Figure 15 ijms-21-07574-f015:**
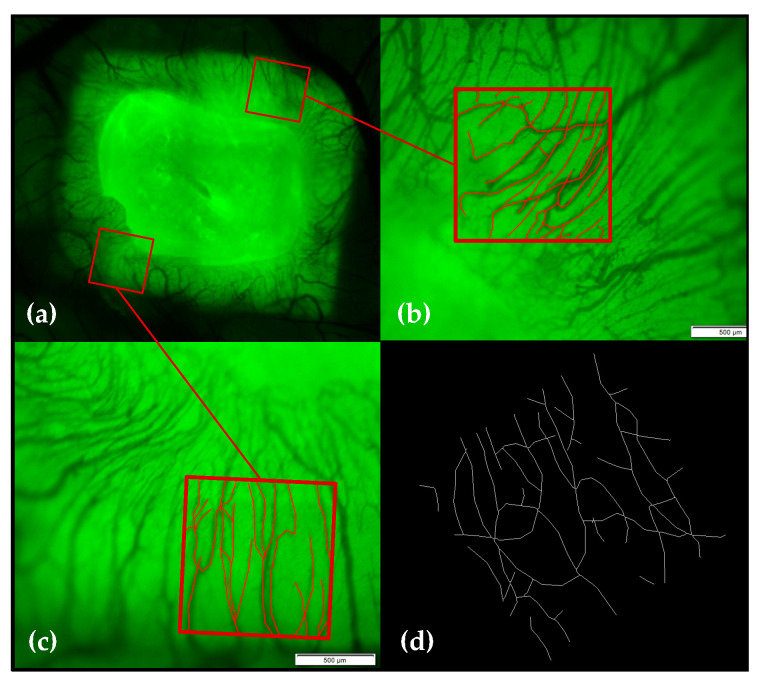
Vessel density and vessel junction analysis. (**a**) The regions of interest (ROIs) were marked within the multiple image array demonstrating the two measured regions chosen within this study. (**b**,**c**) show the grid 1000 × 1000 µm in size laid over the image. All vessels visible inside of the square were traced. The software package automatically calculated the length of the marked lines and the area of the ROI. (**d**) The vessel junction analysis was performed using ImageJ. Images were saved in JPEG image format and further edited. Upon receipt of a suitable image form, the structures were skeletonized; the program automatically calculated the number of vessel branches and junctions per image. The scale bar corresponds to a length of 2 mm.

**Table 1 ijms-21-07574-t001:** Number of specimens analyzed per group.

	Group	Intravital Fluorescence Microscopy	qPCR	Immunohistochemical Staining
**Day 10**	−ESWT	6	NA	NA
	+ESWT	8	NA	NA
**Day 11**	−ESWT	8	8	NA
	+ESWT	9	5	NA
**Day 14**	−ESWT	8	7	4
	+ESWT	9	5	5

**Table 2 ijms-21-07574-t002:** Primers used for quantitative polymerase chain reaction.

Gene	Primer Name	Sequence
**Beta actin gene**	cACTB-s	ACCCCAAAGCCAACAGA
	cACTB-as	CCAGAGTCCATCACAATACC
**HIF-1α**	cHIF1a-s	GCAGCTACTACATCACTTTCTT
	cHIF1a-as	CAGCAGTCTACATGCTAAATCA
**VEGF**	cVEGFA-s	CCTGGAAGTCTACGAACGCA
	cVEGFA-as	CACAGTGAAAGCTGGGTGGT
**MMP-9**	cMMP9-s cMMP9-as	GTCCAGACAGTGGACAAGGG
		CTGGTAACGTGGGGTCATCC
**MMP-13**	cMMP13-s cMMP9-as	CAGGTTTTTCTGGCGACTGC
		TGGGGTAGCCTGTGTCCATA
**iNOS2**	cNOS2-s cNOS2-as	GGCATGATGAGACCCGTAGG
		GCCCAATAGCCACCTTCAGT

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
