# Peer review of "In Vivo Modulation of Angiogenesis and Immune Response on a Collagen Matrix via Extracorporeal Shockwaves"

_ijms, 2020, doi:10.3390/ijms21207574_

Round 1

Reviewer 1 Report

Heimes et al. present an interesting study examining the effect of ESWT on the modulation of angiogenesis and immune response in CM grafting on a CAM assay. The work is elegantly structured and is well designed. The results, however, are not as strong as expected, mostly, I believe, for causes related with low n counts and running only duplicates in RT-PCR tests. The major problem with this work is that most of the results are not significant. Especially in the RT-PCR results, the variance is overwhelming and tremendously limits what we can learn from the results. This is unfortunate as I’m sure that much more could come from such assay. In the end, only vascular density, MMP-9 at day 11 and MMP-13 at day 14 were significantly changed from sham-treated. Looking through the other results, we can see many other changes that show up as mere trends, and from which we can’t make conclusions, that give further hints as to the results produced.

While revising the manuscript the following major and minor comments should be considered:

  1. The authors should reconsider furthering the n-count of this experiment or repeating the RT-PCRs. In the absence of furthering the n-counts of this experiments and clearing the variance at the cost of further sample analysis, the conclusions should be reworded. The observed trends are exciting, but we can’t talk about them as definitive results until they reach statistical significance. This should be made clearer in the conclusions

  1. The results from the vessel junction analysis should be shown. As it is, we only see the graphical representation of the result but not the result itself.

  1. Since the vascular density results show low levels of variance, suggesting that the replicates are good enough, can the authors advance any suggestion as to why the RT-PCR results display such high levels of variance. Also, it’s strange that in some genes, the variance is much higher at day 11 than day 14 and in other genes, the variance is much higher at day 14 than day 11. Were the samples different for different genes?

  1. In line 363, the authors can’t say that the delay could be caused by the time the blood vessels need to enter the tissue as in controls that is not a problem.

Reviewer 2 Report

D. Heimes et all. have studied the impact that extracorporeal shockwaves may have on the tissue regeneration process around a xenogeneic collagen membrane integrated on a CAM model, which provides a refined alternative for animal research. To do so, the authors have evaluated the angiogenesis and immune response during the integration of the biomaterial on the CAM.

The study is well planned, and the data is clearly presented and discussed. Despite the lack of significant differences between the -ESWT and +ESWT groups, the authors have reached to valid conclusions. This, as the author claimed in the discussion section, can be related to the lack of a complete animal physiology system (vasculature, inflammatory and immune system) of the CAM model. Therefore, the authors are contributing with this study to address crucial aspects of the preclinical research-testing pipeline of the CAM model. Some minor comments are:

  • The aim of the study goes more in the direction of assessing the self-healing capability of the model rather than the biocompatibility of the biomaterial. However, the election of the CM does not look arbitrary. The authors missed the opportunity to explain why they decided to combine ESWT with CM and not with other type of biomaterial. It would be very helpful to clarify this in the discussion section.
  • The success of an implant can be measured by the integration of the implant, the presence of a primitive inflammatory response or the survival rate of the chick embryo. In fact, the authors claimed that “a considerable number of eggs dropped out during the experiments.” What was the survival rate at the experimental endpoint for both -ESWT and +ESWT groups? Is it in accordance with the found initial immune response?
  • The election of the number of impulses was based on previous studies. Can different ESWT schedules (i.e. using less impulses or having the same total number of impulses spread in different days) be safer for the normal development of the chick? Would your research direction go this way? A brief discussion regarding this in the last paragraph of the discussion section is highly recommended. The authors could alternatively make this last paragraph a conclusion section.

Round 2

Reviewer 1 Report

No further comments.